

# Harnessing YouTube in advancing biodiversity conservation efforts and awareness across Africa

Anthony Basooma[1,2], Herbert Nakiyende[2], Vianny Natugonza[3] and Rose Basooma[3]

[1] Biodiversity Analytics and Ecohydrology, Asterlook International Ltd, Jinja, Eastern, Uganda
[2] Capture Fisheries and Biodiversity Conservation, National Fisheries Resources Research Institute, Jinja, Eastern, Uganda
[3] Blue Economy and Maritime Studies, Busitema University Institute, Namasagali, Uganda

## ABSTRACT

YouTube (YT), an online video content application, has revolutionized the dissemination of information in various fields, including medicine, entertainment, and conservation science. Its potential in disseminating biodiversity conservation information has not been well assessed, especially in Africa, a biodiversity hotspot. When such assessments are conducted, they are usually species-specific, activity-based, or localized, making broader generalizations difficult. We evaluated the viewership and interaction of the videos posted about Africa across different organism groups, countries (geopolitical units), geographical regions, ecoregions, and channels (content creators). We assessed 431 videos, which collectively garnered 32,630,126 views, 364,700 likes, and 48,839 comments across 274 channels, five regions, and 31 African countries. The mean daily views varied significantly across countries, regions organism groups, and channels. Notably, more views per day were from biodiversity-rich countries, such as Madagascar, and most videos were posted from South Africa. Government and academic institutions posted fewer videos than individually owned and international biodiversity-related non-governmental organization (NGO) channels. Also, most channels posting about African biodiversity are based in the United States of America. Increased attention to biodiversity through social media engagement will likely stimulate external funding, international collaborations, policymaking, and participatory programs, benefiting both the public and organizations such as NGOs and governments. We observed a great need for increased participation by African-based content creators, primarily by government institutions, to effectively adopt a strong social media strategy to communicate information about biodiversity conservation, especially to the growing online population.

# INTRODUCTION

Human-mediated activities have increasingly altered ecosystems and created novel environments (*Hobbs, Higgs & Harris, 2009*; *Lurgi, López & Montoya, 2012*), which do not

Corresponding author
Anthony Basooma,
bas4ster@gmail.com

favor species migration, recruitment, and reproduction (*Hobbs, Higgs & Harris, 2009*). This has escalated species extinction risks, related to the estimated one million plants and animals threatened with extinction (*IPBES, 2019*; *Tollefson, 2019*). This planetary biodiversity crisis due to unsustainable human activities (*Correia et al., 2021*; *World Wildlife Fund, 2022*) has made information dissemination on the potential solutions to alleviate anthropogenic pressures, such as pollution and invasion of alien species, indispensable (*Correia et al., 2021*).

In today's digital or information age, social media platforms, such as YouTube, Facebook, X, and Instagram, are crucial in promoting environmental sustainability, raising awareness of biodiversity conservation-related issues, and understanding the interaction of humans with nature (*Pearson et al., 2016*; *Correia et al., 2021*; *Chowdhury et al., 2024*). However, it can also promote negative activities that affect biodiversity, such as the exotic pet trade (*Moloney et al., 2021*) and animal intolerance, such as wolves and sharks (*Casola et al., 2020*; *Beall et al., 2023*). Social media-derived data, for instance, species occurrence data, is also part of a growing field called 'digital conservation' or 'conservation culturomics' (*Correia et al., 2021*). The data can be used to improve spatial conservation planning and policy action to protect biodiversity (*Freund et al., 2021*; *Chowdhury et al., 2024*)

Much as several dissemination platforms exist, YouTube (YT), an online video-sharing tool, dominates audiovisual environmental communication (*Pavelle & Wilkinson, 2020*). Audiovisual or AV communication combines sound, images, graphics, animation, and interactivity in a presentation tailored for the audience's benefit regarding information, motivation, persuasion, or entertainment (https://proavdc. com/what-is-audio-visual-communication/#:~:text=Audio%20visual%20or%20AV% 20communication%20combines%20sound%2C%20images%2C,audience%E2%80% 99s%20benefit%20regarding%20information%2C%20motivation%2C%20persuasion% 2C%20or%20entertainment). YT gained popularity among young people (*Pavelle & Wilkinson, 2020*), primarily the millennials and Gen Z (*Statista, 2023*). YT is the second-largest search engine and visited website behind Google (*Foster, 2020*; *Statista, 2023*), with more than 2.7 billion monthly users watching approximately one billion hours daily globally (*Shewale, 2023*). In 2022, the platform received approximately 72 billion visits globally (*Statista, 2023*) and was more popular with millennials and GEN-Z, accounting for 52.4% of the users (*Global Media Insight, 2023*). YT allows end users to interact with the content through sharing, commenting, liking, subscribing, viewing, and disliking (*Khan, 2017*). These metrics can be used to explore the extent of information reach, sentiments, and emotions arising after interacting with the information (*Petersen-Wagner & Lee Ludvigsen, 2022*).

YT is among the social media platforms that generate data vital to conservation science (*Toivonen et al., 2019*; *Correia et al., 2021*). YT produces a wealth of user engagement data, such as views, likes, shares, comments, and demographic information, which can help researchers understand public interest in specific species, conservation issues, or environmental campaigns (*Toivonen et al., 2019*). For example, analyzing the viewership and engagement on videos about endangered species or habitat restoration efforts can

provide insights into which topics resonate most with the public, helping shape future conservation messaging and outreach strategies (*Correia et al., 2021*).

Data quality may often be compromised, but YT as a data source is still necessary, especially in regions with low biodiversity funding, mainly in Africa (*Lindsey et al., 2020*). If tapped, YT content can enhance government transparency (*Chatfield & Brajawidagda, 2013*) and develop policies and management strategies to aid in slowing down biodiversity loss (*McBeth et al., 2012*). Conservation science-related studies have highlighted YT's importance, including understanding the illegal wildlife killings (*El Bizri et al., 2015*), and the spread of zoonotic diseases (*Otsuka & Yamakoshi, 2020*). By documenting poaching, wildlife trafficking, and zoonotic outbreaks, YT helps track trends, share educational content, and reach a global audience, fostering better understanding and action about biodiversity conservation. Additionally, YT has been used in wildlife conservation (*Sangi et al., 2024*), assessing the behavioral ecology of wild birds and squirrels (*Jagiello, Dyderski & Dylewski, 2019*; *Casola et al., 2020*), marketing of pets (*Measey et al., 2019*), but also to detect trafficking of threatened species (*Harrington, Macdonald & D'Cruze, 2019*). YT generated data has aided in drawing ecological and social insights into recreational fisheries (*Sbragaglia et al., 2021*) and determining the effects of drones on wildlife (*Rebolo-Ifrán, Grilli & Lambertucci, 2019*). YT has also been used to communicate climate change and global warming, where most comments analyzed were science-related (*Shapiro & Park, 2015*), mostly posted by climate change activists (*Shapiro & Park, 2018*).

Most studies have evaluated the significance of YT in biodiversity conservation and are often based on activity, locality, species, or channel (*Vins, Aldecoa & Hines, 2022*). Local or low-scale studies are informative, and their significance is critical in biodiversity conservation (*Sulis et al., 2021*). However, regional-to-global assessments are equally important as they provide a rapid snapshot of the regions or taxonomic groups that require conservation prioritization. Also, ecological stressors, such as alien species invasion or climate change, have cross-border effects. Thus, localized analysis may not completely used to understand the extent of the effects due to anthropogenic pressures. Since direct data collection for large-scale analyses may be challenging, open-access platforms like YT serve as a critical data source for biodiversity conservation.

Our study focuses on the continental assessment of biodiversity conservation information dissemination across Africa's ecoregions, countries, organism groups, and channels. Within the Global South, conservation funding is continuously less prioritized, making open-access data sources a more probable option to generate data and information. To highlight the reach and extent of biodiversity conservation in Africa, we addressed three hypotheses: (1) the number of views per day, likes per day, and comments per day for videos posted from different geographical regions (*e.g.*, Central Africa, Northern Africa, Western Africa, Eastern Africa and Southern Africa), countries (geopolitical units), and or ecoregions in Africa significantly differ, (2) Viewership and interaction significantly differs across organism groups, with certain taxa, such as mammals or specific wild birds, attracting higher engagement than others. (3) The number of videos posted and viewers that watched significantly differed across different content providers categorized as individually owned, international non-government organizations, and international media outlets.

## MATERIALS AND METHODS

### Data search from YouTube

YouTube video searches were conducted between 26.05 and 02.06.2021 and between 29.09 to 03.10.2023. To put the search in context and determine the scope of work, biodiversity conservation was defined as the genes, species, and all ecosystems, including the human activities that affect these three aspects (*Pimm, 2021*). Ecological processes, such as extinction, define species loss from a particular ecosystem, and therefore, they were considered. We only considered videos concerned with biodiversity conservation from Africa; thus, we included the string "*Africa*" in every search. No country (geopolitical units), geographical region (such as Central, Western, Eastern, Southern, or Northern Africa), or ecoregion was considered directly in the search to avoid geographical or ecological bias.

The search was systemically organized into six categories. These included (1) Ecological processes, which included videos on species taxonomy, ecological, behavioral, or evolutionary processes such as extinction, evolution, and migration. (2) Ecological stressors: This examines the effects, causes, or information on anthropogenic pressures that affect biodiversity. These included videos on non-native species, pollution, climate change, habitat degradation, and water abstraction. (3) Conservation measures and process: This category focuses on conservation strategies such as videos on Ramsar sites, national parks, zoos, protected areas, and nature-based solutions from Africa. (4) Capacity building and training, which considers community empowerment in biodiversity conservation, funding strategies, and the blue economy. (5) Higher-level organism groups: Here, we included organism groups in the search strings, which included "reptiles," "amphibians," "mammals," "birds," "fishes," "plants," and "insects" (Table 1). Finally, the general category included general search terms, including "biodiversity conservation," "conservation," and "biodiversity." This was done to maximize the discovery of more videos on biodiversity conservation (Table 1).

To handle duplicates in the search results across the different categories, we created a dataset where all search results were stored (archived on Figshare: 10.6084/m9.figshare.27969864). Only a video was analyzed if the title, date posted, content provider, and duration differed. This was because the number of views, likes, and comments varied. Videos that provided general information without reference to Africa in the title, content provided, tags, and description were also not considered by this study. Videos about Africa as a continent, but did not refer to any country or region, were labeled as "general" in the country and regional classifications. We also reevaluated the data to identify clickbait videos that offer deceptive, captivating video titles, descriptions, and thumbnails to attract views (*Gothankar, Di Troia & Stamp, 2022*). For example, the video '*The Last Living Dinosaur Could Be Hiding in The Congo*' posted by TheRichest on 23.02.2020 which had garnered 3,342,622.00 views by 19.05.2021 during initial data collection, was identified as clickbait and removed from the analysis dataset (archived on Figshare: 10.6084/m9.figshare.27969864). The original video has been subsequently removed from YouTube.

**Table 1  Summarised search terms.** Search categories, keywords, and video category explanations, including example titles retrieved from YouTube. Overlap in the video returned may exist among categories, but the categories allow the obtaining of videos from diverse biodiversity fields and topics.

| Search categories | Search keywords | Video explanation and examples | Number of videos |
|---|---|---|---|
| Capacity building and training | Biodiversity Assessment, Biodiversity conservation funding; Biodiversity funding; Biodiversity, wildlife and conservation; Blue Economy | These included videos describing empowering conservation strategies through funding and capacity building by organizations such as the United Nations. For example: '*Biodiversity finance landscape in Zambia*'. | 29 |
| Conservation measures | Biodiversity offset; Biodiversity, wildlife and conservation; protected areas such as national parks | Information on biodiversity conservation strategies or facilities, such as protected areas, national parks, Ramsar sites, and breeding grounds. Video example: '*Protected Areas Management in Africa 1.3*' | 17 |
| High organism groups | Amphibians; Bees; Birds; Fish; Insects; Mammals; Plants; Reptiles | Videos returned when particular high organisms were searched. For example, '*Grumpy Looking Rain Frog and GIANT Bug! - Herping Africa*'. | 85 |
| Ecological processes | Evolution; Extinction; Migration | Videos on the species' behavior or processes, such as migration and extinction. Videos, such as '*Amazing wildlife spectacle, the Great Migration, underway in East Africa*'. Since taxa extinction is considered, videos on dinosaurs were found, such as '*The Last Living Dinosaur Could Be Hiding in The Congo*' | 52 |
| Ecological stressors | Climate change; Habitat loss; Invasive species; Overexploitation; Pollution; illegal trade | Videos concerning anthropogenic stressors, including causes, solutions, and intensity. *e.g.*, The '*Most Toxic City in Africa*' | 142 |
| General aspects | Biodiversity conservation; conservation; biodiversity | The videos appeared when a general search '*biodiversity conservation in Africa*'. However, they overlapped the other themes since YouTube identified the videos. For example, '*Biodiversity of Madagascar*' | 106 |

## Extracting data metrics from the videos

The number of views, likes, comments, video title, duration, date posted and retrieved, channel name (content creator), channel country of origin (where it was registered), and number of subscribers were retrieved. Since we searched for higher-level organism groups, such as mammals, reptiles, and amphibians, we further watched to identify organism groups with lower-level organism groups. For example, in videos concerning mammals, we further identified whether it's about elephants, lions, or humans. Two criteria were followed to extract the taxon: criteria (1) indication of the taxon in the video title on which the content was based; (2) the presence of animals or parts of the animal in the video content, whether in the forest or markets, such as pangolin scales or elephant tusks because of illegal trade. We maintained simplified identification, such as humans, elephants, and lions, to ensure no expert knowledge is required for species identification. This aspect allows non-technical researchers in species identification to conduct a coarse taxonomic analysis of biodiversity information.

Some video content was general and addressed many biodiversity aspects without a particular emphasis on the taxon, environmental process, threat, or region within

Africa, and they were categorized as "general". All countries were grouped into geographical areas based on the African Union classification (https://au.int/en/member_states/countryprofiles2).

To identify the ecoregions, we obtained country centroids from (https://github.com/360-info/country-centroids) and identified where they fall within the ecoregion polygon using **st_join** in the sf package (*Pebesma, 2018*). The regions were extracted from the Regional Center for Mapping of Resources for Development (RCMRD) (https://opendata.rcmrd.org/datasets/africa-ecoregions/about). Ecoregions are areas with distinct species assemblages and ecological conditions (*Olson et al., 2001*).

The channels that posted videos were classified into nine categories: (1) individually owned (registered on the YT with an individual name); (2) academic institutions (*i.e.,* universities or institutions of learning, whether local or international). (3) International media outlets (*i.e.,* media outlets with coverage beyond Africa, such as BBC, Al Jazeera, CTGN Africa, and DW Documentary). These included channels exclusive to Africa but under a conglomerate, such as BBC Africa and CTGN Africa. (4) International non-government organizations (NGO), which included biodiversity-related international non-profit making organizations such as the National Geographic Society (NGS), International for Conservation of Nature (IUCN), Flora and Fauna International, BirdLife International, and World Wildlife Fund. (5) International or continental institutions that included organization that have political governance such as the World Bank, United Nations, African Development Bank, and associated entities such as Convention on Biological Diversity Secretariat, IPBES Secretariat and UNEP. (6) Local organizations, including NGOs and organizations operating at the country level but not Government or media outlets. (7) Government institutions (ministries, departments, and research institutions). (8) Regional institutions within a particular region but not continental, such as Freshwater Research, which operates in South African countries. (9) Local media outlets which operate within the country (such as NTV Uganda). To determine if channels are based in Africa, after classifying them, we collated their country of origin from the 'About' section on the YT or searched on individual websites. We classified the information as NI (not identified) (archived on Figshare: 10.6084/m9.figshare.27969864) if the information about the YT channel was unavailable.

To cater for differences in the number of views or likes based on the variation in the time spent on YT, we standardized the number of views, comments, and likes. We calculated the number of days a video has spent on YT as the difference between the date retrieved and the date posted on YT. This was computed using the *time_length* and *interval* functions from the *lubridate* package (*Grolemund & Wickham, 2011*). The number of views, likes, and comments was then standardized by dividing by the number of days the video has been on YT.

**Table 2  Mean number of views per day, likes per day, and comments per day for videos.** Mean number of views per day (vpd (±standard error)), likes per day (lpd (±SE)), and comments per day (cpd (±SE)) for videos posted from regions of Africa. The videos in Africa do not address any particular region but the continent as a whole. N is the total number of individuals from each region. Kruskal Wallis tests the Chi-squared value ($\chi^2$), degree of freedom (DF), and p-value for difference among the regions.

| Regions | N | vpd (± SE) | lpd (± SE) | cpd (± SE) |
|---|---|---|---|---|
| Africa | 142 | 133.16 ± 48.18 | 6.24 ± 4.56 | 0.62 ± 0.37 |
| Central Africa | 19 | 100.92 ± 64.81 | 0.82 ± 0.32 | 0.48 ± 0.42 |
| East Africa | 111 | 225.23 ± 102.46 | 3 ± 1.3 | 0.4 ± 0.23 |
| North Africa | 15 | 3.63 ± 1.9 | 0.1 ± 0.07 | 0.02 ± 0.01 |
| South Africa | 96 | 116.93 ± 54.9 | 1.15 ± 0.49 | 0.19 ± 0.06 |
| West Africa | 48 | 61.12 ± 27.86 | 0.82 ± 0.35 | 0.21 ± 0.1 |
| ($\chi^2$) | | 9.54 | 8.35 | 8.07 |
| DF | | 5 | 5 | 5 |
| p-value | | 0.089 | 0.1377 | 0.1526 |

## Statistical analysis to evaluate the differences in views, likes, and comments across regions, countries, taxa, ecoregions, and channel categories

We assessed variation in the number of views per day (vpd), comments per day (cpd), and likes per day (lpd) across regions, countries, taxa, and channel categories. Before comparing differences among the countries, geopolitical regions, ecoregions, content creator categories, higher-level organism groups, and lower-level organism groups, we tested if the sample data was generated from a normal distribution using the Shapiro test. Also, the homogeneity of variance among groups was evaluated using the Levene test. If the two assumptions were violated, then non-parametric one-way ranked Kruskal–Wallis instead of one-way ANOVA was applied. We then used the Dunn *post hoc* tests to determine the statistical differences between groups if significant differences were observed. During statistical analysis, groups with fewer than three videos were grouped as ''others''.

## RESULTS

### Biodiversity information reach and interaction across regions, countries, taxa, and channel categories

We collated 431 videos, generating 32,630,126 views, 364,700 likes, and 48,839 comments from 31 African countries. The video views accumulated in 79 h and 264,714,291 subscribers from 274 channels. Most videos (32.8%) considered the whole continent, followed by those focused on Eastern Africa at 25.6%, and the least from Northern Africa (Table 2). The views per day (vpd), likes per day (lpd), and comments per day (cpd) were highest in Eastern Africa and least in Northern Africa. The differences among the regions were not statistically significant (Table 2).

Only 252 videos (58.3%) were from individual countries. Most videos were specific to South Africa, followed by Madagascar, Uganda, Kenya, and Guinea-Bissau (Fig. 1). Madagascar had the highest mean vpd, lpd, and cpd (Fig. 2). The vpd was generally higher than the likes and comments per day. The vpd and lpd were significantly different among

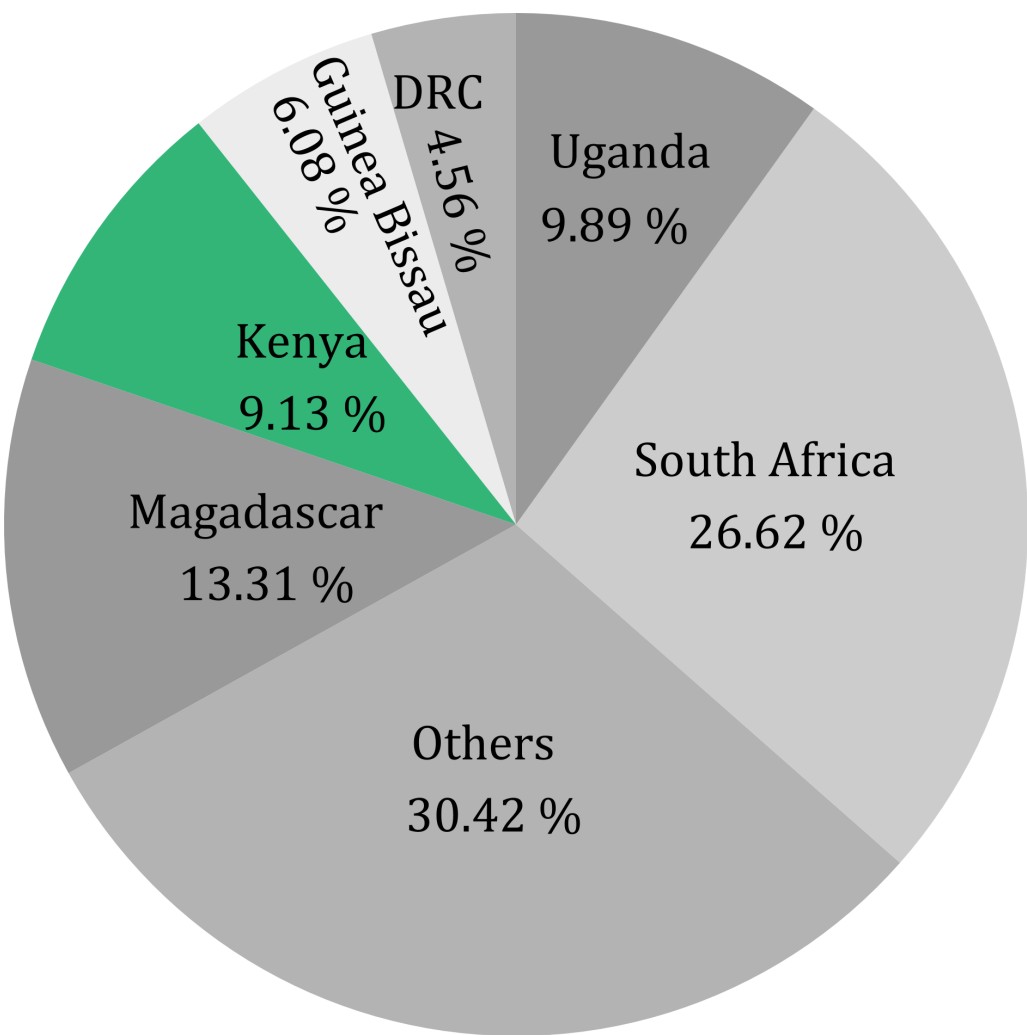

**Figure 1** **Total number of videos posted from each country.** These were compared for videos that were based on a specific country. South Africa was only included as the country, not as a region. Countries with fewer than ten videos were categorized as others.

countries (vpd: Kruskal–Wallis $\chi^2 = 32.78$, $df = 13$, $p$-value = 0.001) and (lpd: Kruskal–Wallis $\chi^2 = 27.59$, $df = 13$, $p$-value = 0.010), but not cpd (Kruskal–Wallis $\chi^2 = 16.096$, $df = 13$, $p$-value = 0.24). *Post hoc* Dunn test showed that Madagascar had significantly higher numbers of vpd and lpd compared to other countries ($p < 0.001$) (Table S1).

Based on ecoregions, the highest number of videos were posted in the Kalahari xeric savanna (78), followed by Madagascar subhumid forests (35), Victoria Basin forest-savanna (33), and the least were in the Sudd flooded grasslands, Mediterranean dry woodlands, and steppe, Western Guinean lowland forests which had only one video. In contrast, the highest mean views per day were recorded in the Madagascar subhumid forests (743.8), followed by the Sudd flooded grasslands (204.2) and the Guinean forest-savanna at 148.1 (Fig. 3).

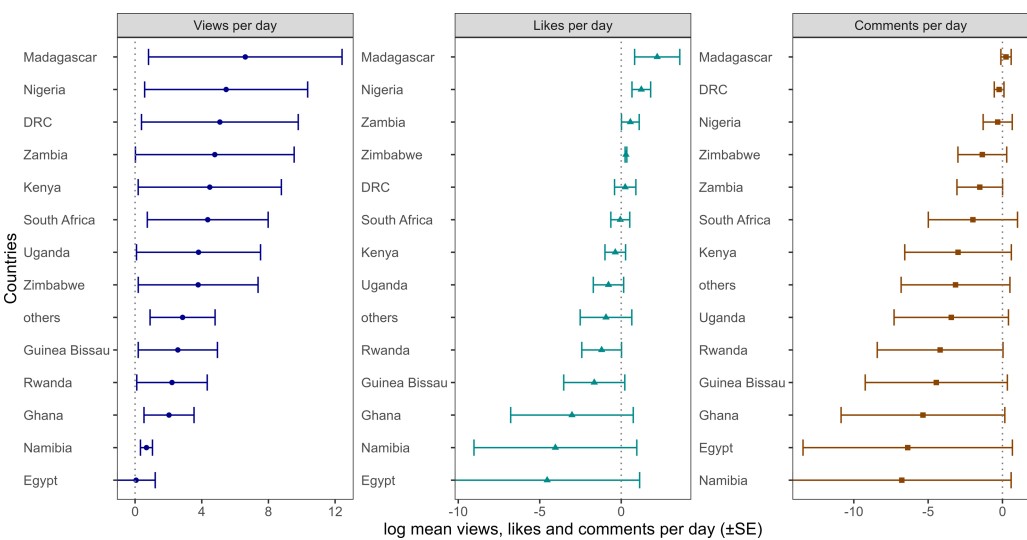

**Figure 2  Video views, likes and comments per day from different African countries.** Total number of videos, mean views per day, likes per day, and comments per day across African countries. The mean values were log-transformed to visualize even the small numbers. Only countries in Africa with at least five videos were visualized.

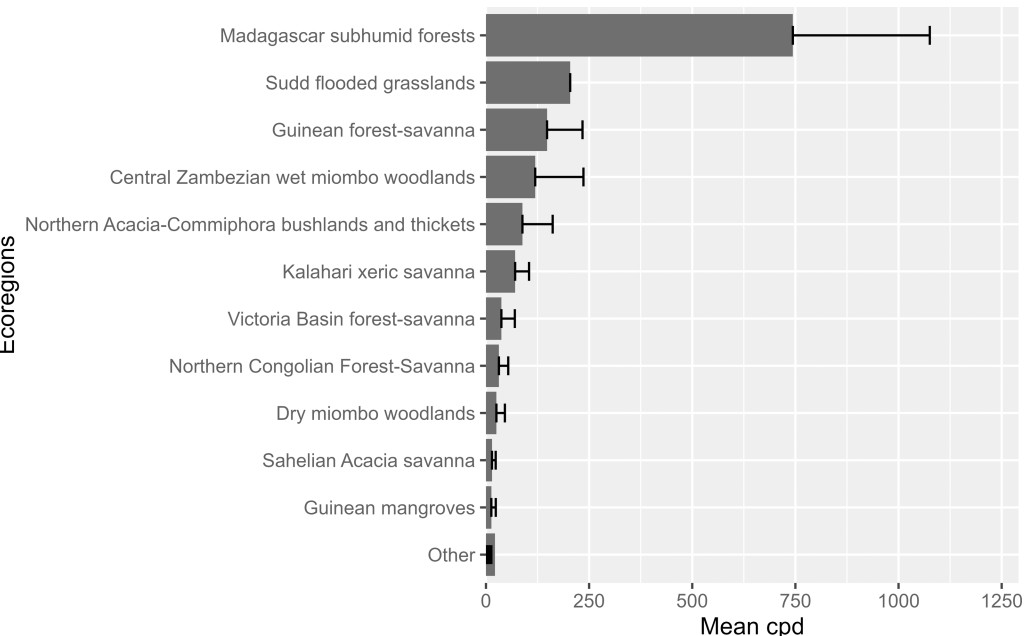

**Figure 3  Videos posted from African ecoregions.** Mean views per day for the ecoregions found in Africa.

Based on the higher-level organism groups, 171 videos were obtained, dominated by mammals at 88 videos (51.4%) and lowest for insects, at three videos (1.7%) (Table 3). Fishes and mammals had the highest mean video vpd, lpd, and cpd (Table 3). Both vpd
**Table 3 Mean number of views per day (vpd), likes per day (lpd), and comments per day (±SE) for videos posted for higher-level organism groups.** N is the total number of videos obtained for each taxonomic group. Kruskal–Wallis test ($\chi^2$), degree of freedom (DF), and pvalue for difference among higher organism groups.

| Higher-level organism groups | N | vpd (± SE) | lpd (± SE) | cpd (± SE) |
|---|---|---|---|---|
| Amphibians | 11 | 2.006 | 0.025 | 0.005 |
| Birds | 9 | 67.605 | 5.306 | 0.481 |
| Fishes | 30 | 232.892 | 3.599 | 0.836 |
| Insect | 3 | 0.435 | 0.019 | 0.000 |
| Mammals | 88 | 77.166 | 1.268 | 0.123 |
| Plants | 22 | 4.743 | 0.239 | 0.047 |
| Reptiles | 8 | 1.163 | 0.016 | 0.001 |
| ($\chi^2$) | | 17.93 | 15.59 | 14.27 |
| **DF** | | 6 | 6 | 6 |
| *p*-value | | 0.006 | 0.016 | 0.02 |

and lpd significantly differed across the higher-level organism groups (Table 3). A pairwise comparison using the Dunn test showed that insects had significantly fewer views per day (vpd) compared to other taxa ($p < 0.05$) (Table S2).

Among the lower taxonomic groups, fishes (specifically coelacanths) and reptiles (dinosaurs) received the highest number of views (Fig. 4). The Kruskal–Wallis test revealed a significant difference in vpd among lower taxonomic groups ($\chi^2 = 35.12$, $df = 19$, $p$-value $= 0.01$). *Post hoc* Dunn test showed that vpd for coelacanth, fossa, and dinosaurs were considerably higher than other lower organism groups, such as bees, apes, and humans ($p < 0.05$). Both lpd and cpd were not significantly different among the lower organism groups, including elephants, vultures, and apes, cpd (Kruskal–Wallis $\chi^2 = 27.28$, $df = 19$, $p$-value $= 0.09$) and lpd (Kruskal–Wallis $\chi^2 = 23.18$, $df = 19$, $p$-value $= 0.22$).

Out of the 431 videos, individually owned channels were the most dominant at 37.2%, followed by international media outlets at 27.7%, such as BBC News and Al Jazeera English (Table 4). Fewer videos were posted by local media outlets, government institutions, and local organizations (Table 4). The vpd, lpd, and cpd were significantly different among the channel categories (Table 4). The *Post hoc* Dunn test showed that international NGOs, international media outlets, and individual channels had significantly higher vpd, lpd, and cpd (Table 4). Most channels are based in the United States, followed by Kenya, South Africa, and the United Kingdom (Fig. 5).

## DISCUSSION

This study investigated the reach (views) and interaction (likes and comments) of biodiversity-related information on YT across organism groups, geographical regions, ecoregions, countries, and channel categories in Africa. Generally, the findings reveal that viewership and interaction with biodiversity-related videos from Africa are still lower than in other fields, such as disease prevention. For instance, *Diers et al. (2023)* analyzed 130 videos about the usefulness of YT as a source of information about asthma,

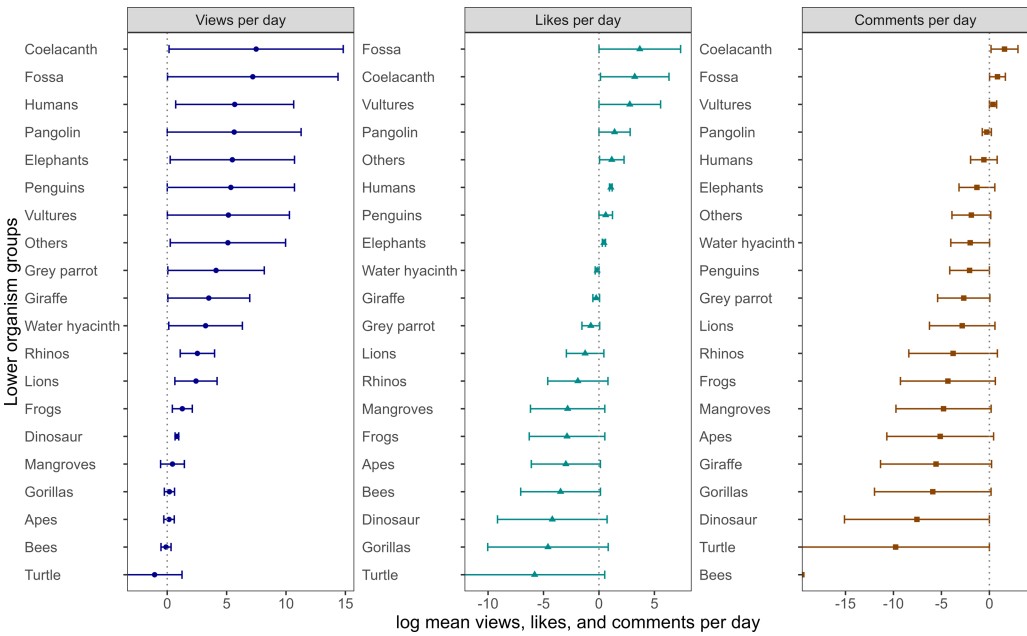

**Figure 4** **Number of videos views, likes and comments per day form different organism groups.** Total number of videos, mean views per day (vpd), likes per day (lpd), and comments per day (cpd) across the lower organism ranks in Africa. We log-transformed the mean values to visualize small values, which led to a negative scale in both the views, likes, and comments per day graphs.

**Table 4** **Mean number of vpd (±standard error), lpd, and cpd for videos by channel category.** N is the total number of videos from each region. Kruskal Wallis Chi-squared ($\chi^2$) value, degree of freedom (DF), and p-value for difference among the channel categories.

| Channel category | N | vpd (± SE) | lpd (± SE) | cpd (± SE) |
|---|---|---|---|---|
| Academic institution | 14 | 10.9 ± 8.35 | 0.08 ± 0.07 | 0.01 ± 0.01 |
| Government | 14 | 1.72 ± 1.04 | 0.01 ± 0.01 | 0 ± 0 |
| Individual | 160 | 271.04 ± 82.52 | 7.6 ± 4.14 | 0.82 ± 0.36 |
| International NGO | 36 | 139.6 ± 65.4 | 1.64 ± 0.8 | 0.3 ± 0.2 |
| International institutions | 42 | 3.78 ± 1.16 | 0.02 ± 0.01 | 0 ± 0 |
| International media outlet | 120 | 92.41 ± 40.17 | 0.88 ± 0.26 | 0.22 ± 0.08 |
| Local media outlet | 14 | 13.67 ± 10.95 | 0.06 ± 0.05 | 0.03 ± 0.03 |
| Local organization | 15 | 1.33 ± 0.48 | 0.03 ± 0.02 | 0 ± 0 |
| Regional Institution | 16 | 0.8 ± 0.23 | 0.02 ± 0.01 | 0 ± 0 |
| ($\chi 2$) | | 40.44 | 46.08 | 51.13 |
| DF | | 8 | 8 | 8 |
| *p* | | 0.000002 | 0.0000002 | 0.00000002 |

which garnered 29.8 h and accumulated 100,290,242 views compared to 79 h from this study, but with only 32,630,126 views. Evaluating the usefulness and accuracy of only 60 COVID-19-related videos on YT accumulated 257,804,146 views (*Li Heidi et al., 2020*). In contrast, these studies were compared globally, possibly leading to significant differences in the viewership. Even so, conservation managers and scientists require directed efforts
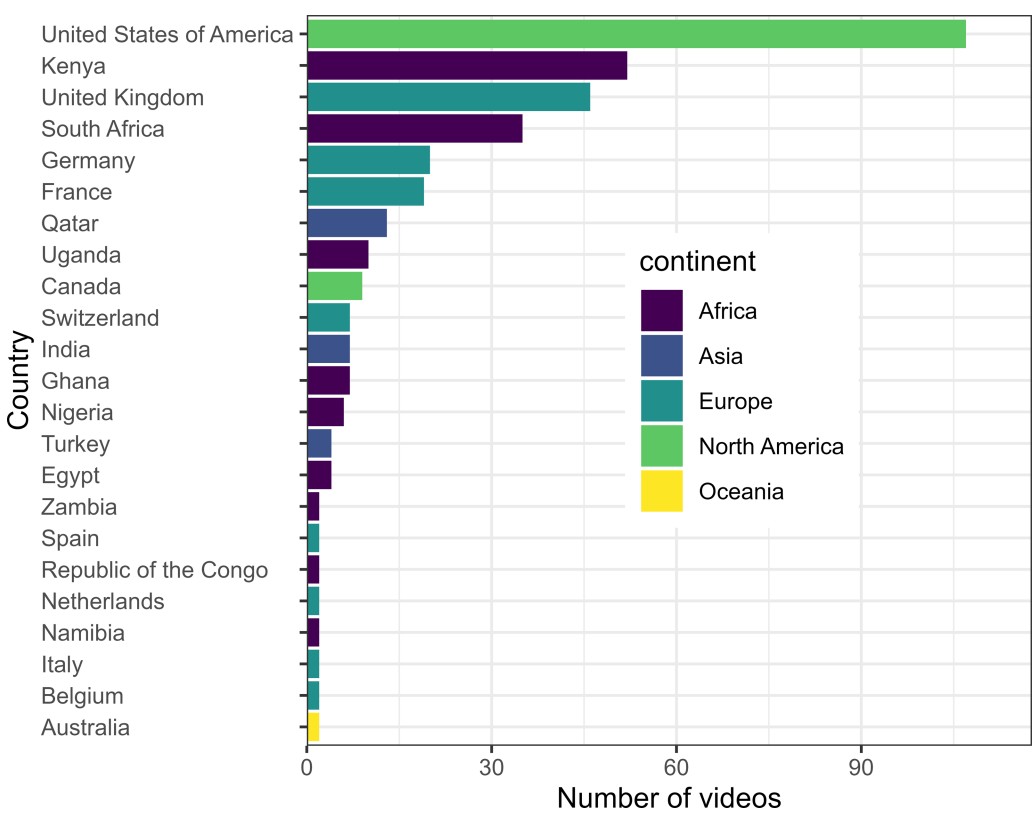

**Figure 5** Content providers/channels and the countries where they are hosted.

to attract the masses to watch and interact with biodiversity conservation videos to steer mindset change.

At a regional level, Eastern Africa had a numerically high number of views, likes, and comments per day, although these differences were not statistically significant. Madagascar, an island known for its unique biodiversity, garnered significantly viewership. The island's endemic species and international research efforts (internationally recognized as a biodiversity hotspot with about 90% of the 13,000 species) likely contribute to its broader appeal to the audience (*Phillipson et al., 2006*). The island is home to the coelacanth, which was thought extinct about 420 million years ago. Still, now it is regarded as the epicenter of the West Indian Ocean coelacanth (*Latimeria chalumnae*) population (*Cooke, Bruton & Ravololoharinjara, 2021*). Therefore, the island attracts international evolutionary and ecological studies, possibly increasing its audience scope. Although all other countries or regions are highly diverse, no particular videos have been posted on YT that attracted a wider viewership. For instance, most of the videos were posted from South Africa but primarily based on water and air pollution from Mpumalanga, of which the topical issue of pollution is a global threat. The haplochromine cichlid diversity and endemicity of Lake Victoria and Lake Malawi have not received any attention except for water hyacinth, a global threat.
For ecoregion level, the mean views per day were higher in the Madagascar subhumid forests. The forests were highly diverse but highly affected by anthropogenic stressors, including deforestation, pollution, poaching, and invasive species. It has attracted significant international attention as it was listed on the WWF's major list of important ecoregions (*Olson et al., 2001*).

At the higher-level organism groups, videos related to insects received significantly fewer views and likes per day, indicating that channel operators have organism preferences. Videos on topics such as coelacanth, fossa, and dinosaurs, which featured unique and captivating evolutionary histories, tended to attract more views. For example, '*Fossa: the King of Madagascar*,' posted by the Animologic channel on 09.07.2018, had garnered 2,770,683.00 by 19.05.2021.

In Fig. 4, the most charismatic animals, such as tigers, lions, and elephants (*Albert, Luque & Courchamp, 2018*), do not top the list, meaning that video content's appeal is not only driven by charisma but by other factors like storytelling, emotional engagement, and striking visuals. While elephants, for example, symbolize grandeur and evoke strong emotions, and species like frogs spark curiosity due to their distinct features, animals like the fossa captivate audiences due to their rarity and mysterious nature. *Albert, Luque & Courchamp (2018)* research on the Western public's perception of charismatic species highlights that charisma is not always synonymous with popularity but also beautiful, impressive, or endangered. Moreover, what is considered attractive or popular in the Global North may differ significantly from perspectives in the Global South, particularly in Africa, where people often coexist with many of these animals and plants in their daily lives. For instance, the fossa, usually described as "not a cat, not a monkey, and not a ferret," stands out in videos due to its uniqueness and role as Madagascar's top predator. The unfamiliarity of such species adds to their intrigue and draws in viewers, much like an uncommon, compelling human story would attract attention. For instance, many human-related videos focused on pollution in South Africa, a well-known and common issue in Africa. Similarly, videos featuring lemurs, arguably Madagascar's most iconic species, attracted fewer views. For example, a video posted for 211 days (https://www.youtube.com/watch?v=uAvfYtZ4aRU) garnered only 1,302 views. This may be due to the prominence of other species or shifting media focus towards less well-known animals, like the fossa, which is gaining attention for its unique characteristics.

The videos based on insects received fewer viewership and were also less posted. Despite contributing about 80% of the animal species (*Stork, 2018*), they are often neglected in conservation assessments, policies, and frameworks, such as the African Union's 1968 Convention on the Conservation of Nature and Natural Resources (*African Union, 2003*). This neglect is reflected in social media content, which affects the support for implementing their conservation efforts.

YouTube videos play a significant role in shaping public views on species and conservation. Previous studies have shown that YT videos can influence perceptions and tolerance towards wildlife, as seen in studies on wolves (*Casola et al., 2020*) and sharks (*Beall et al., 2023*). *Fidino, Herr & Magle (2018)* emphasize how online opinions and social media platforms, including YouTube, shape public perceptions of wildlife. YouTube

provides a platform to raise awareness, with videos reaching large audiences and fostering discussions on critical conservation issues. For example, *Moloney et al. (2021)* found that videos featuring threatened exotic animals can promote positive and negative perceptions, depending on how the content is presented. This highlights the potential for YouTube to educate viewers and influence conservation behavior, driving support for conservation efforts.

Mostly individual channels, international media outlets (*e.g.*, Al Jazeera and BBC News), and international biodiversity-related non-government organizations, such as National Geographic, generated more views and interactions than government and international institution-operated channels. This was similarly observed when government and healthcare professionals posted only 6.87% and 5.47% of videos on the COVID-19 pandemic compared to 71.63% by news channels (*Parabhoi et al., 2021*). *Diers et al. (2023)* also reported that TV shows posted 26.6% of the videos on asthma compared to only 7% posted by lung specialists. Furthermore, professional healthcare providers posted 24% compared to 76% by News channels (*Remvig et al., 2022*). The reluctance of government institutions and professionals to post videos on YouTube or social media has led to the spread of false information. The reason for curtailing professional bodies or government institutions from posting YT videos may be the cost of filming and editing the YT content. For example, a relatively cheap and basic video project may cost $2,000 or $3,000 to $10,000 for middle-level projects (*Henriksen, 2004*). These costs may be inhibiting factors for most professional institutions, mainly in the Global South, that prefer low-cost information dissemination mediums such as reports and posters. However, similar to *Akyol Onder & Ertan (2022)*, we advocate for the most reliable information sources, such as government institutions and professional bodies, to use YT to disseminate biodiversity information to the masses. These can include do-it-yourself (DIY) short clips about biodiversity that would not require high production costs.

While all videos analyzed are from Africa, a significant portion of them are created by international (non-African) content creators, with most channels based in the United States, with some contributions from Kenya, primarily from CTGN Africa, an international news agency. This raises a key issue: YouTube, rather than solely disseminating information, often plays a role in creating narratives around conservation. Unfortunately, when external content providers dominate the conversation narrative, it risks sidelining the perspectives and expertise of local African communities. This imbalance can undermine the authenticity of conservation messages and highlight the need for greater involvement of African voices in shaping the conservation agenda and disseminating biodiversity information. Also, to decolonize biodiversity conservation science, more locally rooted or indigenous-based approaches are required to shift from mainstream conventional measures such as protected areas (*Büscher & Fletcher, 2020*; *Corbera et al., 2021*). Also, local content creators should drive the dissemination of biodiversity conservation information across social media networks, potentially reducing the spread of false information.

### Limitations of the study

Generally, the search algorithm YT is not straightforward, so we had less influence on the search results. This affects the study's reproducibility since if a different user searches using the same key terms, different search results may appear. Secondly, although no country was specified in the search terms, YT contents from some countries may have been restricted, which made it undiscoverable in the search results. For example, YT was banned in Ethiopia in 2023 and in most North African countries, such as Morocco, Algeria, Libya, and Tunisia (*Robertson, 2020*). Third, since only videos in English were considered, possibly videos generated in local languages, if present, were not considered. Fourth, most local institutions concerned with biodiversity usually have fewer videos and do not appear in the searches. Therefore, we recommend searches from channels at the local level to understand if the biodiversity information reaches the target audience fully. Fifth, since the ecoregions are usually shared among multiple countries or multiple ecoregions shared in one country, this may affect the accurate allocation of the videos assessed. Finally, YouTube lacks quality control, and the accuracy of the information can be questionable, such as posting clickbait videos (*Gothankar, Di Troia & Stamp, 2022*). Therefore, fact-checking and oversight are needed to ensure reliable conservation messages and prevent the spread of misinformation.

## CONCLUSIONS

This study highlights the potential for YT as a platform for disseminating and creating biodiversity information across organism groups, ecoregions, countries, and content creators. It further underscores the importance of reliable information sources, the unique appeal of regions with high biodiversity, and the role of captivating and educational content in attracting audiences to biodiversity-related videos. YT has fully attracted viewership, mainly to regions, countries, and ecoregions with high biodiversity in Africa, potentially facilitating resource mobilization from external funding sources. However, significant effort is still required by professional bodies and government institutions to embrace technology and disseminate information to a broader populace. Increasing the visibility of insects and other organisms on platforms like YouTube could raise awareness and drive more robust conservation actions across Africa. This insight is vital for conservation managers and scientists to foster collaborative efforts to address these challenges on a continental scale.

### Funding

The authors received no funding for this work.

### Competing Interests

The authors declare there are no competing interests.

## Author Contributions

- Anthony Basooma conceived and designed the experiments, performed the experiments, analyzed the data, prepared figures and/or tables, authored or reviewed drafts of the article, and approved the final draft.
- Herbert Nakiyende performed the experiments, authored or reviewed drafts of the article, and approved the final draft.
- Vianny Natugonza performed the experiments, authored or reviewed drafts of the article, and approved the final draft.
- Rose Basooma performed the experiments, analyzed the data, prepared figures and/or tables, authored or reviewed drafts of the article, and approved the final draft.

## Data Availability

The data is available at figshare: Basooma, Anthony; Nakiyende, Herbert; NATUGONZA, VIANNY; Basooma, Rose (2025). Evaluating the potential of YouTube in disseminating biodiversity-related information from Africa. figshare. Dataset. https://doi.org/10.6084/m9.figshare.27969864.v2.

## Supplemental Information

Supplemental information for this article can be found online at http://dx.doi.org/10.7717/peerj.19545#supplemental-information.

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
