# Peer review of "Harnessing YouTube in advancing biodiversity conservation efforts and awareness across Africa"

_PeerJ, doi:10.7717/peerj.19545_

## Round 0.1 · original submission · Major Revisions

I enclose two particularly detailed, thoughtful reviews. Both reviewers are in favour of your manuscript. Both provide a lot of specific changes. You're so lucky.! We've all had reviews that take for ever and make no practical suggestios.

So, please revise and send item-by-item responses to the comments. I will certainly send it back to the reviewers to ensure that they feel their comments have been addressed.

Reviewer 1 ·

Basic reporting

I want to congratulate the authors for conducting a nice and important study on the importance of harnessing YouTube (YT) videos to enhance biodiversity study and conservation in Africa. The data collected and used are very important, and an addition to the study of emerging digital platforms for biodiversity conservation, especially, as organisms respond to environmental change. This article will fit well into PeerJt’s publication criteria and the study warrants publication after some major revisions are made. Specifically, I have some major concerns regarding the methods used. However, I think these are addressable in the context of the study’s key hypotheses. I also have some minor suggestions in both the introduction, results and discussion sections, and I have them highlighted in the included document.

Experimental design

Lines 102-103: How did you choose which "one" video to include? It would have been great if you used a scoring and ranking procedure to avoid excluding the most important video. The way it is mentioned here, it appears there was just an arbitrary removal of videos. I would like you to explain the established scientific procedures used in including or excluding videos. I would recommend authors to adopt the approaches established (e.g., https://onlinelibrary.wiley.com/doi/10.1111/ddi.13771; https://conbio.onlinelibrary.wiley.com/doi/10.1111/conl.12985)
Lines 165-169: I feel like the statistical analyses used to address the stated hypotheses are very basic and might not test these very important hypotheses properly (see https://link.springer.com/article/10.1007/s11160-022-09699-7 on how to use GLMs to analyse data from social media platform, such as YouTube). The Post hoc analyses test hypotheses best when they are used on the outputs of models (see; https://www.sciencedirect.com/science/article/pii/S2405844024011629?via%3Dihub).

Validity of the findings

Lines 199-201: The interpretations were a bit confusing. Would you consider rephrasing to make the message clearer?
Lines 213-219: I struggled to see the relevance of these to the overall objectives or a single objective of the study. I would probably move these to supplementary information to keep my results tight and interesting to the readers.

Additional comments

Minor comments.
Introduction
Lines 38-40: Seem way off the opening paragraph. I will recommend deleting that while replacing the sentence with how we can address this negative trend (here, you can introduce how social media data from people are being used to address this concern conservation and species migration issues; see https://conbio.onlinelibrary.wiley.com/doi/10.1111/cobi.14161 and https://ecoevorxiv.org/repository/view/7660/)
Lines 54-68: These two paragraphs can be one paragraph to improve the readability of this study.
Line 84: I would think that at the first mention, you will mention all the regions in Africa (i.e., North, South, East, West and Central Africa). This is because you would want your reader to have a good understanding of your study area. Mentioning “Africa” as your study and constantly having only East and South Africa throughout your manuscript is not a good representation of the region. And, not everyone who will read your paper is conversant with the geography of the region.

Materials and methods
Lines 92-93: How many records were returned from these searches? How did you screen your records to meet your criteria for inclusion in the final data? See my comment above for further details and recommendations.

Annotated reviews are not available for download in order to protect the identity of reviewers who chose to remain anonymous.

Reviewer 2 ·

Basic reporting

Basic reporting is adequate, and paper is generally well written. Additional references would help. See additional comments for more details.

Experimental design

Research design is adequate, though the study is purely descriptive. I suggest removing the content analysis and word cloud, as that seems very subjective and doesn't add much to the paper. See additional comments for more details.

Validity of the findings

More concrete implications of the work could be discussed and clarified. See additional comments for more details.

Additional comments

This is an intriguing paper focused on an increasingly important issue – the role of social media in biodiversity conservation. The paper is largely descriptive and doesn’t offer many concrete, practical insights for conservation. Nevertheless, given its novelty and potential ability to advance the conversation about social media in conservation, I still believe the manuscript could be published pending major revisions. Some of the major issues that should be addressed include the following:

1. In the Intro, the authors could provide more examples about the ways in which YouTube can inform and/or influence conservation outcomes. They mention “data,” but what kinds of data can YouTube provide? How might YouTube videos actually influence (a) people’s views of species/conservation and (b) conservation efforts themselves? Below, I have listed some references that might help as the authors elaborate on these effects. But expanding in this area is critical to help readers understand the potential impacts of YouTube videos.

2. In the Methods, the authors need to say more about how this research was conducted. That should include a comprehensive list of all keywords used, more details about the coding scheme, etc. Perhaps provide a table that defines each code and links to at least one video as an example. It might help to include all of this in supplemental information (perhaps it was there already, but I couldn’t find it). This information is needed in order for the study to be replicable. I also found it odd that certain videos like the one about the lost dinosaur made it through filters. How is that video relevant to contemporary biodiversity conservation? More details on methods could help readers understand this.

3. I suggest deleting the objective (#4) about ecological stressors and all of the results associated with it (e.g., the word cloud in Fig. 6). I don’t find this content analysis section, or the word cloud associated with it, to be particularly useful. It sounds arbitrary (completely contingent on keywords, and it’s not clear how/why these were selected) and appears to be rather piecemeal… I don’t see how any meaningful conclusions could be drawn from this. If the authors want to retain the word cloud, they need to justify its use as an analytical tool. Perhaps it could be moved to supplemental files?

4. A key finding in this paper is the fact that most of the videos about African conservation did NOT come from Africa. This (i.e., the fact that outside voices are driving the conversation) speaks to larger issues of colonialism in conservation. I urge the authors to say much more about this, and its long-term implications, in the Discussion. I have suggested some references below that should help on that front.

5. In the Discussion, the authors should also say more about what makes YouTube videos so appealing/captivating. What attributes lead to a successful video that attracts people’s attention? For instance, many of the most liked videos did not involve classic charismatic species like tigers, lions, elephants, etc (see a suggested reference below about that). What was it about the videos on obscure species (e.g., fossa) that captured attention? It would be helpful for readers to know what qualities of YT videos are likely to yield the best responses, as this could generate a more impactful conservation response. A little more speculation about this is warranted.

In addition to these larger recommendations, I have also included line-by-line edits and suggestions for the authors to consider below. I hope the authors find these suggestions helpful should they choose to revise their manuscript.

Abstract:
L13 – If what is conducted? Evaluation? Clarify.
L15 – Rather than mentioning the particular value in reaching young people here, you might shift this up to the beginning of the abstract (L11). I say this because your study did not specifically examine the effects of age (not in your data), but it widely known that younger people are more likely to engage with social media.

Introduction:
L36-40 – I find the specific focus on freshwater ecosystems here to be somewhat odd. It makes it sound like the entire paper will focus on freshwater ecosystems, but that is not the case. Can you downplay this and/or list other types of ecosystems that might be threatened?
L44 – Can you say a little more about what you mean by “audiovisual environmental communication?” Communication about what? Maybe provide an example or two (ranging from X to Y).
L45 – Font for the Pavelle references appears to be different than other text.
L47 – Is this global users? Clarify geographic scope.
L54 – What kinds of data does it generate that might be useful to conservation science? Can you provide an example or two? Data about, or related to, what?
L60-66 – Related to previous comment, can you say more about how YT informs these things? What, exactly, does it do to help people understand illegal killings, zoonotic diseases, etc.
L64 – Font for the Otsuka reference appears different again, please check to make sure fonts aren’t switching. I’ll stop pointing this out now.
L67 – How has YT been used to evaluation response to climate change? Provide a brief example.
L69 – Say “Most studies that have evaluated…”, then delete the (“, but…”) later in the sentence.
L71 – “is not contestable” is strange wording here. Could you just say they are critical/important?
L83 – Key word here is FROM. Are these videos from these regions (i.e., posted in that region – how would you know?), or videos about these regions (i.e., with country keyword in the title)? Very important distinctions. Based on our analysis, it sounds like the latter, not the former, since much of the content originated from outside Africa.
L87-88 – What do you mean by “channel categories” here? Can you provide a parenthetical example or two?
L88 – The concept of “ecological stressors” was not adequately discussed in the Intro. Define what this means earlier, perhaps when describing threatened ecosystems (and why they are threatened). However, I find this aspect of the analysis to be a little more arbitrary and less compelling, so you might consider deleting the objective all together.

Materials and Methods:
L96 – Are these keywords in parentheses (extinction, evolution) just examples, or an exhaustive list of all terms that were used? If exhaustive, aren’t there many more ecological processes that might included (e.g., pollination). How were these terms selected? Can you say more about that, or provide a full list of keywords considered in the search?
L100 – How did you determine if a video was related to biodiversity conservation? What were the criteria for this decision? If a study like this is to be replicable, more details are needed in methods to help readers see how these decisions were made.
L103 – I don’t really understand this line. Do you mean that duplicates were deleted? Clarify.
L115 – Returning to a previous comment, you need to provide more info about your coding process and categories (including definitions, etc.). I suggest doing this in a supplemental file, where codes for each category are described in more detail, perhaps with links to example videos that matched that code. This information might already be in the submission, but I couldn’t find it.
L117 – What did you do if a large country contained multiple ecoregions, which is often the case? The centroid might fall into one, but a major chunk of land in the country might fall into another. Explain how you dealt with this potential scenario (or note it as a limitation).
L161 – Although a word cloud provides an interesting overview of terms that appear in the videos, it’s not really an analysis. Can you say more about why you chose to use the word cloud approach and how it helps you understand the biodiversity information present in the videos? Perhaps referencing an article like this would help:
DePaolo, C. A., & Wilkinson, K. (2014). Get Your Head into the Clouds: Using Word Clouds for Analyzing Qualitative Assessment Data. TechTrends, 58(3), 39.
L166 – This is a very large sample… which makes me wonder why you used non-parametric tests (e.g., Kruskal Wallis) to analyze data. Can you explain your choice of non-parametric stats here, instead of something like ANOVA?

Results:
L196 – I was very surprised to see that reptiles had the highest mean scores. Why was this? Was it driven by just 1-2 outlier videos? Perhaps you should note this here.
L202 – Why would these species in Madagascar generate so much attention relative to others? Hopefully you speculate about this in the Discussion.
L202 – Dinosaurs?! I thought this study was about contemporary biodiversity conservation. In that case, how did dinosaur videos get through the filter? In what ways are the relevant today?
L202 – You say “lower” organisms here, but are humans “lower” than fish? Maybe just delete this word here and in the next sentence (L204).
L213-219 – As noted earlier, I don’t find this content analysis section, or the word cloud associated with it, to be particularly useful. I recommend deleting it, or perhaps only including the word cloud as a supplemental file.

Discussion:
L225 – Clarify that the Diers study was about disease.
L241 – Wording is off here. The phrase “viewership and interaction” doesn’t fit the rest of the sentence. Reword.
L242-243 – The second part of this sentence feels redundant and unnecessary. Delete or reword.
L250 – I was surprised to see that Madagascar was a popular place, yet there were no videos about lemurs (probably the most popular species on the island). Why might that be? Perhaps this should be noted.
L251 – Change the word “peculiar” to particular
L255 – Not sure if “menace” is the right word here? Threat?
L256 – This comment about the subhumid forests in Madagascar makes me wonder about the correlation between country and ecoregion. If the ecoregion was determined based on the country centroid, wouldn’t they be highlight collinear? Is this is a problem?
L263-266 – This text about the coelacanth is not a complete a sentence. Reword.
L266 – As noted above, how is this video about hidden dinosaurs related to contemporary biodiversity issues? It sounds like sensational clickbait. Should the dinosaur video be included in the analysis?
L271 – Again, I find the use of the word “peculiarity” here to be… peculiar. I suggest picking a different word. I’m not sure what you mean here.
L276 – Does social media content affect implementation itself, or just support for various implementation efforts? Clarify.
L282 – Citation format is off here (sentence starts with parentheses).
L286 – This example about orthodontics feels odds and out of place. Consider deleting.
L297 – I think more is needed here in the Discussion (and perhaps in the Intro as well), about how YT videos might actually influence (a) people’s views of species/conservation, and (b) conservation efforts themselves. To make these points, you need references. I suggest consulting previously published papers on this topic, perhaps starting with these:
Fidino, M., Herr, S. W., & Magle, S. B. (2018). Assessing online opinions of wildlife through social media. Human Dimensions of Wildlife, 23(5), 482-490.
Casola, W. R., Rushing, J., Futch, S., Vayer, V., Lawson, D. F., Cavalieri, M. J., ... & Peterson, M. N. (2020). How do YouTube videos impact tolerance of wolves?. Human Dimensions of Wildlife, 25(6), 531-543.
Moloney, G. K., Tuke, J., Dal Grande, E., Nielsen, T., & Chaber, A. L. (2021). Is YouTube promoting the exotic pet trade? Analysis of the global public perception of popular YouTube videos featuring threatened exotic animals. PLoS One, 16(4), e0235451.
Beall, J. M., Pharr, L. D., von Furstenberg, R., Barber, A., Casola, W. R., Vaughn, A., ... & Larson, L. R. (2023). The influence of YouTube videos on human tolerance of sharks. Animal Conservation, 26(2), 154-164.
L298 – Can you say more here about the implications of international actors driving the African conservation agenda on social media. It speaks to larger issues of colonialism in conservation. Maybe reference papers like these (there are many more on this topic too):
Agrawal, A. (1997). The politics of development and conservation: legacies of colonialism. Peace & Change, 22(4), 463-482.
Domínguez, L., & Luoma, C. (2020). Decolonising conservation policy: How colonial land and conservation ideologies persist and perpetuate indigenous injustices at the expense of the environment. Land, 9(3), 65.
L300 – This goes back to one of my earlier points. Were these videos from Africa or about Africa. Also, is YT really about disseminating information or creating new narratives? You might want to make that distinction here. It seems like what we have is outsiders telling the story (or setting the agenda) for African conservation – and that’s not really a good thing.
L302-315 – I don’t think this section adds much, and I suggest deleting it. It sounds very piecemeal and arbitrary, and there is little effort – or even ability – to synthesize with a larger body of literature because the text-based analysis is relatively meaningless.
L317 – Need to say more here about the limitations of the search terms that were used. And perhaps more about the coding process too (e.g., did you have multiple raters to confirm inter-rate reliability of codes)?
L322 – What do you mean by video quality here?

Conclusions:
L327 – It would be great to say more about the appeal of YT videos in the discussion (i.e., their captivating quality). That is what makes YT so powerful. In the Discussion, can you provide example or at least offer potential explanations (citing relevant literature) as to why certain YT videos were more captivating than others? It would be helpful for readers to know what qualities of YT videos are likely to yield the best responses, as this could generate a more impactful conservation response.
L332 – I think it has to be mentioned here that there is no quality control on YT, and the accuracy of the information presented is often questionable. I would like to see a caveat here noting that some degree of oversight and fact-checking may be needed to avoid spreading misinformation.
L335 – I suggest reiterating one final point here – the fact that very few of these videos came from Africa. Remind readers that it’s important to elevate local voices to avoid colonial conservation.

Figure 1 – Videos from each country, or about each country? See my notes about this earlier.
Figure 4 – Looking at this figure, I am surprised to see that the most charismatic animals (tigers, lions, elephants) are not topping the list. That is surprising to me. How might you explain this? In the Discussion, I would say more about why this is the case. Perhaps explain what makes video content appealing, and note that it doesn’t have to involve a charismatic species (it could be a fossa). And be sure to cite this paper when talking about this discrepancy:
Albert, C., Luque, G. M., & Courchamp, F. (2018). The twenty most charismatic species. PloS one, 13(7), e0199149.
Figure 6 – As noted above, I suggest deleting the word cloud and maybe moving it to supplemental files.
Table 1 – To clarify that the first row (Africa) is a sum of the others, maybe add (total) after Africa.
Table 2 – Title says Table 1. Be sure to fix this. Also, this highlights an earlier point I made – the reptile scores are surprisingly high. Is there 1 video driving this and skewing results? Be sure to note that in the Results section of the paper.

---

## Round 0.2 · Minor Revisions

I have only been able to get one re-review of your manuscript. Please address the minor comments the reviewer requests and resubmit. I expect to be able to make a decision to accept quickly.

Reviewer 1 ·

Basic reporting

The revised manuscript addressed all my concerns about the previous version of the manuscript. The revised manuscript now reads better, with no confusing statements to make the readability difficult. The literature used in the revised manuscript is also very relevant and up-to-date in the context of this research. I have no further comments on the basic reporting.

Experimental design

Although I do not agree with the statistical approach used and the justifications given by the authors in their response letter. I would have loved to see a rigorous statistical method, considering that skewed data can still be handled by the Poisson (log-link) family in a GLMM. Thus, I would give it to them that their study design is sufficient in the context of their aim and scope. More importantly, the additional information on the search terms makes the study design very easy to replicate. I commend the authors for taking the time to improve the method section of this revised manuscript. I have no further comment on the experimental design.

Validity of the findings

The findings are very valid, with great potential to extend our understanding of how social media, like YouTube usage is shaping conservation science.

Additional comments

My minor comments on this revised manuscript would be to recommend to authors to have a thorough read of their manuscript and correct some font sizes, types, typos, punctuation, and inconsistencies in their in-text references. Just to highlight a few;

Line 48 has Correia et al. 2021, which is different from Correia et al., 2021 in line 42
Line 52: The reference need a space between the , and the year 2022
Line 78: add comma before "and"

---

## Round 0.3 · accepted · Accept

Thanks for taking care of these minor comments so quickly. And for submitting your interesting paper to PeerJ!